# Plant and animal functional diversity drive mutualistic network assembly across an elevational gradient

Jörg Albrecht[1], Alice Classen[2], Maximilian G.R. Vollstädt[1], Antonia Mayr[2], Neduvoto P. Mollel[3,4], David Schellenberger Costa[5,6], Hamadi I. Dulle[1], Markus Fischer[1,3], Andreas Hemp[7], Kim M. Howell[8], Michael Kleyer[5], Thomas Nauss[9], Marcell K. Peters[2], Marco Tschapka[10,11], Ingolf Steffan-Dewenter[2], Katrin Böhning-Gaese[1,12] & Matthias Schleuning [1]

Species' functional traits set the blueprint for pair-wise interactions in ecological networks. Yet, it is unknown to what extent the functional diversity of plant and animal communities controls network assembly along environmental gradients in real-world ecosystems. Here we address this question with a unique dataset of mutualistic bird–fruit, bird–flower and insect–flower interaction networks and associated functional traits of 200 plant and 282 animal species sampled along broad climate and land-use gradients on Mt. Kilimanjaro. We show that plant functional diversity is mainly limited by precipitation, while animal functional diversity is primarily limited by temperature. Furthermore, shifts in plant and animal functional diversity along the elevational gradient control the niche breadth and partitioning of the respective other trophic level. These findings reveal that climatic constraints on the functional diversity of either plants or animals determine the relative importance of bottom-up and top-down control in plant–animal interaction networks.

[1] Senckenberg Biodiversity and Climate Research Centre (BiK-F), Senckenberganlage 25, 60325 Frankfurt am Main, Germany. [2] Department of Animal Ecology and Tropical Biology, Biocenter, University of Würzburg, 97074Am Hubland, Würzburg, Germany. [3] Institute of Plant Sciences, University of Bern, Altenbergrain 21, Bern 3013, Switzerland. [4] Tropical Pesticides Research Institute (TPRI), Arusha, Tanzania. [5] Landscape Ecology Group, Institute of Biology and Environmental Sciences, University of Oldenburg, Carl von Ossietzky Straße 9-11, 26129 Oldenburg, Germany. [6] Institute of Ecology and Evolution, Friedrich Schiller University Jena, Dornburger Strasse 159, 07743 Jena, Germany. [7] Department of Plant Systematics, University of Bayreuth, Universitätsstraße 30, 95440 Bayreuth, Germany. [8] Department of Zoology and Wildlife Conservation, University of Dar-es-Salaam, Dar-es-Salaam, Tanzania. [9] Environmental Informatics, Faculty of Geography, University of Marburg, Deutschhausstraße 12, 35032 Marburg, Germany. [10] Institute for Evolutionary Ecology and Conservation Genomics, University of Ulm, Albert- Einstein-Allee 11, 89069 Ulm, Germany. [11] Smithsonian Tropical Research Institute, PO Box 0843-03092Balboa Ancòn, Republic of Panama. [12] Institute for Ecology, Evolution and Diversity, Goethe University Frankfurt, Biologicum, Max-von-Laue-Straße 13, 60439 Frankfurt am Main, Germany. Correspondence and requests for materials should be addressed to J.A. (email: joerg.albrecht@senckenberg.de)

All species are involved in mutualistic and antagonistic interactions with other species[1]. Collectively these interactions between pairs of species form complex networks that structure ecological communities[2] and maintain essential ecosystem functions, such as pollination, seed dispersal or biological control[3]. Within these networks the matching of species' traits determines whether pairs of species are able to interact and how effective their interactions are[4]. Therefore, phenotypic traits related to species' interactions are thought to determine the realised set of interactions within species-rich interaction networks[4]. In a broader sense, the functional traits that regulate species interactions can be viewed as coexistence traits that govern niche breadth (the diversity of a species' interaction partners) and niche partitioning (the complementary specialisation of several species on exclusive interaction partners) in complex ecological networks and determine the ecosystem functions derived from ecological communities[4–6].

In theory, we would expect that species' niche breadth and the degree of niche partitioning among species, increases with functional diversity[7,8]. If trait matching structures plant–animal interaction networks, the functional trait spaces of plants and animals should reciprocally control the niche breadth and partitioning of species in the respective other trophic level (Fig. 1a–c). Hence, a reduction of functional diversity in the lower trophic level is expected to cause a reduction in niche breadth and partitioning (niche contraction and convergence) in the higher trophic level and vice versa (bottom-up and top-down control, respectively; Fig. 1d). The convergence of interaction niches also implies that a reduction of functional diversity in one trophic level may cause increased competition for mutualistic partners in the other trophic level. The functional diversity in one trophic level may thus not only constrain the interaction niches but also the functional diversity in the other trophic level through biotic filtering and competitive exclusion[9].

Our framework implies that bottom-up and top-down effects may simultaneously control the assembly of plant–animal interaction networks and that abiotic constraints on either plant or animal functional diversity may determine the relative importance of bottom-up or top-down control[10]. This prediction goes beyond those from biodiversity experiments, as the latter have mainly focussed on either bottom-up or top-down effects, but have not yet assessed the relative importance and context dependence of both mechanisms simultaneously[10–12]. Despite a general consensus on the relevance of functional diversity for species interactions and associated ecosystem functions in real-world ecosystems[13–15], it is unknown to what extent shifts in plant and animal functional diversity along environmental gradients alter the structure of species interaction networks and whether network assembly is primarily bottom-up-controlled or top-down-controlled.

Here, we ask whether trait matching is a general phenomenon across mutualistic networks and whether bottom-up and top-down forces simultaneously control the assembly of these networks in real-world ecosystems. To address these two questions, we recorded a unique dataset of mutualistic bird–fruit, bird–flower and insect–flower interaction networks and associated functional traits along a 3.5 km elevational gradient (872–4396 m above sea level [a.s.l.]) of near-natural and

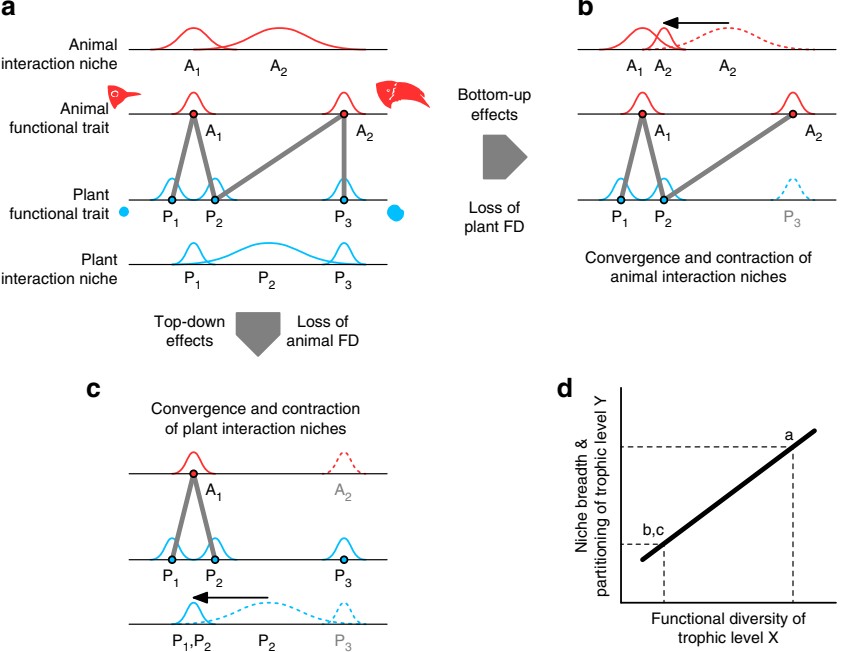

**Fig. 1** Potential bottom-up and top-down effects of plant and animal functional diversity on the assembly of mutualistic networks. **a** Example of a mutualistic bird–fruit network, in which plants and animals are ordered along two corresponding size-matching trait axes according to bill and fruit size, respectively. The grey lines represent bird–fruit interactions, which are constrained by trait matching so that the small-billed bird can only consume small fruits, whereas the large-billed bird can also consume large fruits. Trait matching determines the realised interaction niches of plants and animals (represented on the trait axis of the other trophic level). **b** Removal of plant species $P_3$ causes a loss of plant functional diversity and a contraction and convergence of the birds' interaction niches corresponding to a reduction in niche breadth (diversity of a species' interaction partners) and niche partitioning (complementary specialisation of several species on exclusive interaction partners; red histograms $A_1$ and $A_2$ at the top). **c** Likewise, removal of bird species $A_2$ causes a loss of animal functional diversity and a contraction and convergence of plants' interaction niches (blue histograms $P_1$ and $P_2$ at the bottom). **d** Consequently, a loss of functional diversity in one trophic level should cause a reciprocal reduction in niche breadth and partitioning in the other trophic level. Note that the convergence of interaction niches also implies that a reduction of functional diversity in one trophic level may cause increased competition for mutualistic partners in the other trophic level (e.g., between $A_1$ and $A_2$ in **b** or between $P_1$ and $P_2$ in **c**)

anthropogenic habitat on the southern slopes of Mt. Kilimanjaro. The dataset comprises a total of 14,728 interactions between 200 plant and 282 animal species (99 bird and 183 insect species) sampled across the three types of mutualisms. First, we tested whether trait matching generally structured the assembly of these networks by analysing multivariate associations between plant and animal functional traits. Second, we tested with a Bayesian hierarchical structural equation model to what extent changes in plant and animal functional diversity along the elevational gradient control network assembly.

We find consistent evidence that trait matching determines pair-wise interactions across mutualisms. Plant functional diversity is primarily limited by precipitation, while animal functional diversity is mainly constrained by temperature. We further discover that shifts in plant functional diversity along the elevational gradient control the niche breadth and partitioning of animals in the interaction networks, while animal functional diversity determines interaction niches of plants. Therefore, our findings reveal that environmental constraints on either plant or animal functional diversity determine the relative importance of bottom-up and top-down control in plant–animal interaction networks.

## Results

**Trait-associations in plant–animal mutualistic networks**. To characterise the functional trait spaces of plants and animals in each of the three types of mutualistic networks, we selected traits related to size matching (matching traits), to energy provisioning and requirements (energy traits), as well as to foraging stratum and mobility (foraging traits; Fig. 2 and Methods section)[16-24]. We projected plants and animals in each mutualism into their multidimensional trait spaces and assessed functional relationships between plant and animal trait spaces using a combination of RLQ and fourth-corner analyses, taking into account which combinations of species pairs were observed to interact[24-26] (see Fig. 2 and Methods section).

A null model, in which we randomised species' identities[27], indicated that the global associations between plant and animal trait spaces were larger than expected by chance in the bird–fruit (sum of RLQ-eigenvalues: $\Sigma\lambda_i = 0.18$, $P < 0.01$), and in the insect–flower mutualism ($\Sigma\lambda_i = 0.083$, $P < 0.05$), and tended to be larger than expected in the bird–flower mutualism ($\Sigma\lambda_i = 0.16$, $P = 0.074$). The marginal trend in the bird–flower mutualism is likely due to a lack of statistical power owing to the relatively small number of species (26 plant and 24 bird species, respectively)[27]. The first ordination axis explained most of the cross-covariance between plant and animal trait spaces (range across the three mutualisms: 90–99%), whereas the second axis explained only a minor proportion (range: 0.75–9.3%). In separate analyses of the two ordination axes, associations between the first axes of plant and animal trait spaces were stronger than expected by chance in all three mutualisms (bird–fruit: Pearson's $r = 0.26$, $P < 0.001$; bird–flower: $r = 0.33$, $P < 0.01$; insect–flower: $r = 0.23$, $P < 0.01$). Associations between the second axes were generally weaker and significant only in the bird–fruit mutualism (bird–fruit: $r = 0.13$, $P < 0.05$; bird–flower: $r = 0.040$, $P = 0.67$; insect–flower: $r = 0.063$, $P = 0.40$). Across the three mutualisms, the first axes of plant and animal trait spaces were most strongly correlated with matching traits (absolute Pearson's $|r| = 0.21 \pm 0.016$ [mean ± s.e.m.], Moran's test[28]: $P < 0.001$, see Methods) and energy traits ($|r| = 0.19 \pm 0.024$, $P < 0.001$), whereas correlations with foraging traits were weaker and more variable in magnitude ($|r| = 0.11 \pm 0.034$, $P = 0.031$; Supplementary Table 1; Supplementary Fig. 1). The second axes of plant and animal trait spaces were not correlated with any of the trait types (matching traits: $|r| = 0.035 \pm 0.0074$, Moran's test: $P = 0.74$; energy traits: $|r| =$ 0.031 ± 0.013, $P = 0.74$; foraging traits: $|r| = 0.055 \pm 0.014$, $P = 0.23$; Supplementary Table 1; Supplementary Fig. 1).

**Bottom-up and top-down effects of functional diversity**. We assessed whether shifts in plant and animal functional diversity along gradients of climate and land use drive network assembly in the three mutualisms using Bayesian hierarchical structural equation models with a stochastic variable selection procedure (Fig. 3; Supplementary Fig. 2; see Methods). The structural equation models tested for consistent direct and indirect 'functional diversity'-mediated effects of mean annual temperature, mean annual precipitation and land use on niche breadth and partitioning of plants and animals across the three mutualisms. We quantified niche breadth as the mean effective number of partners based on the exponent of the Shannon diversity of links ($e^H$), and niche partitioning as the mean standardised Kullback–Leibler distance ($d'$) across the plants and animals in each network, respectively[29].

Plant functional diversity was positively related to mean annual precipitation, whereas animal functional diversity increased with mean annual temperature. In line with our prediction (Fig. 1), we found that plant functional diversity was positively associated with niche breadth and partitioning of animals, while animal functional diversity was positively related to niche breadth and partitioning of plants (Fig. 3b, c; Supplementary Table 2; Supplementary Fig. 3). Mean annual temperature was also directly positively associated with the niche breadth of animals, as well as with the niche partitioning of plants and animals (Fig. 3b, c). We found no consistent direct or indirect effects of land use on functional diversity, niche breadth or partitioning of plants and animals across the three mutualisms (Fig. 3b, c).

Structural equation models including univariate functional diversity metrics based on matching, energy or foraging traits showed that the increase in plant functional diversity with mean annual precipitation was primarily driven by an increase in the variability of foraging strata (i.e., plant height), while the increase in animal functional diversity with mean annual temperature was primarily driven by an increase in the variability of matching traits (i.e., bill width, bill length and proboscis length, respectively; Supplementary Fig. 4). Niche breadth and partitioning of plants, as well as niche breadth of animals were related to the functional diversity of all trait types (matching, energy and foraging traits), whereas niche partitioning of animals was mainly related to functional diversity of foraging strata in plant communities. In addition to the multivariate analysis, univariate models indicated an increase in the variability of size-related energy traits in response to land use (i.e., body mass in the bird mutualisms and head width in the insect mutualism; Supplementary Fig. 4).

## Discussion

Our study provides a general assessment of the importance of trait matching and functional diversity for the assembly of mutualistic networks. We show that matching of species' functional traits is a general mechanism regulating interactions in mutualistic networks. Importantly, we discover that plant and animal functional diversity are related to distinct climatic factors and constrain the realised niche breadth and partitioning of the respective other trophic level. Hence, our study reveals that environmental constraints on either plant or animal functional diversity drive the relative importance of bottom-up and top-down effects on mutualistic network assembly.

We found that functional traits related to size matching were strongly associated with network structure across mutualisms, because size matching imposes critical barriers that either directly prevent interactions between plants and animals or strongly

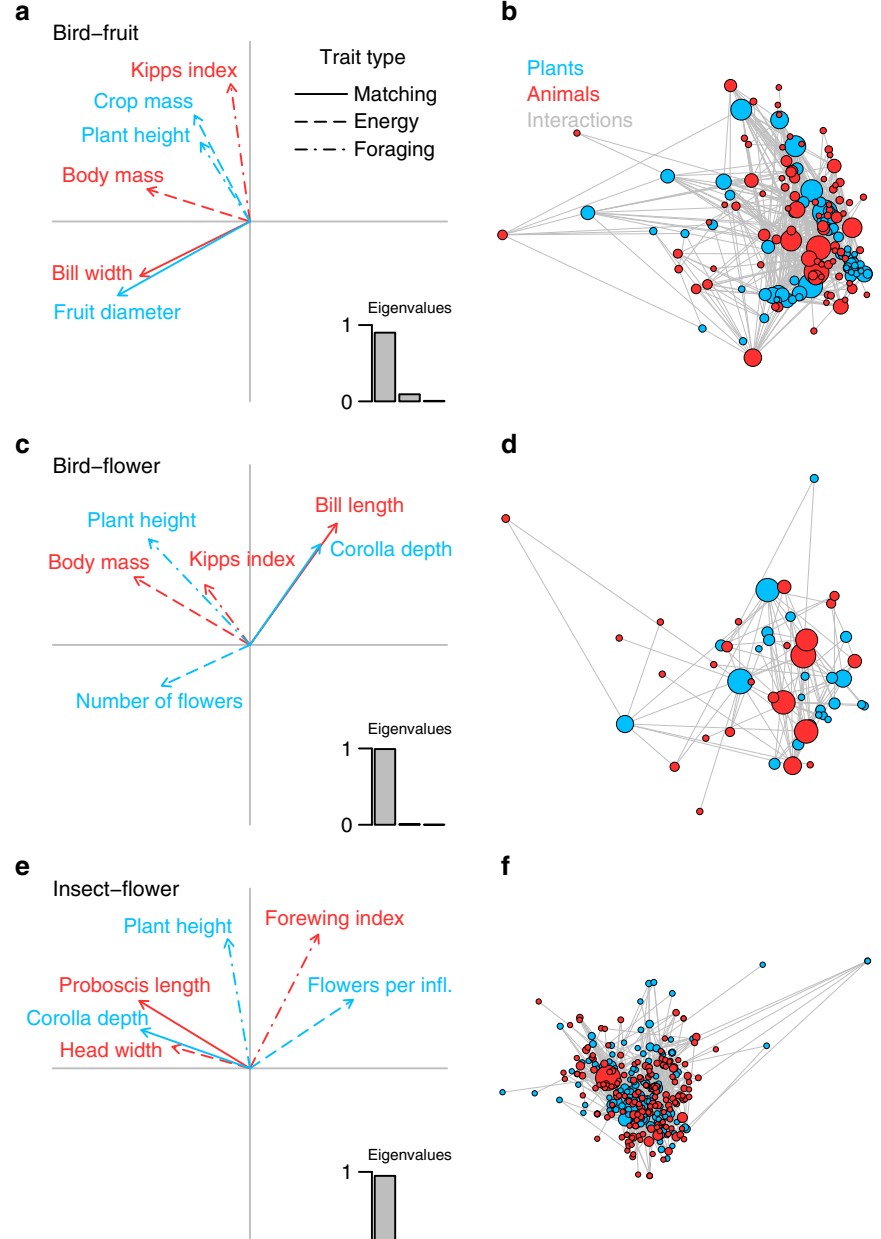

**Fig. 2** Associations of plant and animal functional traits in different types of plant–animal mutualistic networks. Results of a combination of RLQ and fourth-corner analyses for **a**, **b** bird–fruit, **c**, **d** bird–flower, and **e**, **f** insect–flower mutualisms. We use RLQ analysis (**a**, **c**, **e**) to map the multivariate trait space of animals on the multivariate trait space of plants based on plant–animal interaction networks. The eigenvalues of the RLQ analysis in **a**, **c**, **e** indicate the proportion of the cross-covariance between plant and animal trait spaces explained by each RLQ axis[25]. Vectors in **a**, **c**, **e** depict the coefficients of plant (blue) and animal (red) traits on the first two axes of the plant and animal trait spaces from RLQ analysis. If two vectors are long and point into the same or opposite directions the absolute correlation coefficient between the corresponding traits is large. Different line types in **a**, **c**, **e** indicate different types of functional traits, related to size matching (continuous line), energy provisioning of plants and energy requirements of animals (long-dashed line), as well as foraging stratum and mobility (dash-dotted line). **b**, **d**, **f** Representation of the mutualistic networks in the multivariate trait spaces of plants and animals. Dots in **b**, **d**, **f** represent species scores of plants (blue) and animals (red) in their multivariate trait spaces and grey lines depict interactions between plants and animals. The size of the dots is proportional to the number of links that a species has (i.e., species degree)

constrain their effectiveness[17,30]. Traits related to energy provisioning and requirements were also closely associated with network structure, which can be explained by optimal foraging theory[31], because larger animals with higher energetic requirements should prefer spatially clustered resources (i.e., plants with a high resource density) to reduce the energetic costs of foraging[32]. Yet, the direction of the relationships between traits related to energy requirements of animals and resource

provisioning by plants differed between bird and insect mutualisms (Fig. 2a, c, e): while larger birds tended to forage on plants with higher resource density, larger insects foraged on plants with lower resource density. In the insect–flower mutualism matching and energy traits were positively correlated in animals, but negatively correlated in plants (Fig. 2e). These contrasting trait associations in plants and animals suggest that trade-offs with matching traits might alter associations between energy traits in

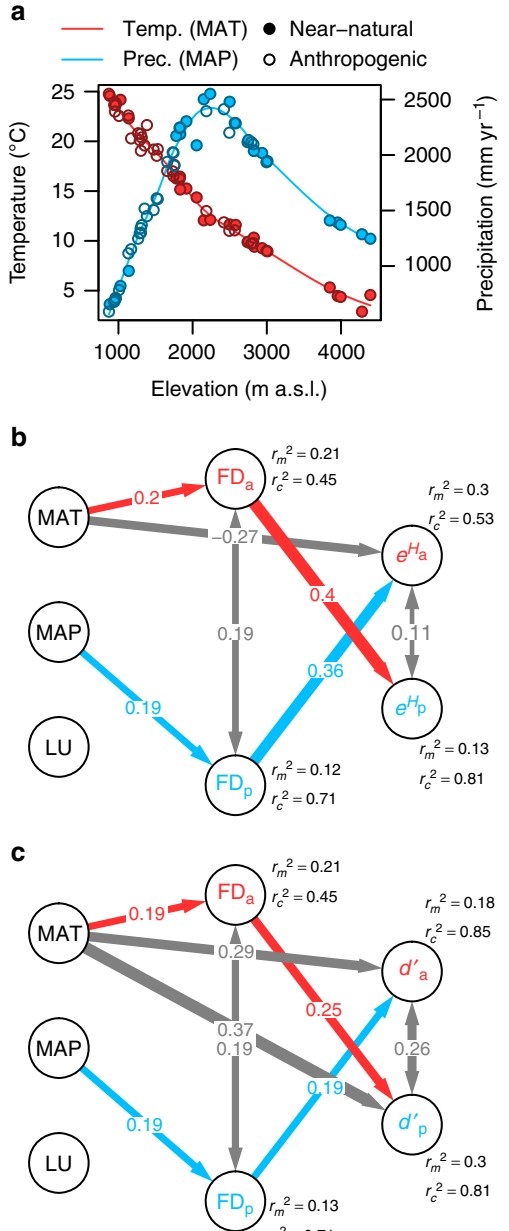

**Fig. 3** Reciprocal bottom-up and top-down effects of functional diversity on the assembly of plant–animal mutualistic networks. **a** Variation in mean annual temperature (MAT [°C], red) and mean annual precipitation (MAP [mm yr$^{-1}$], blue), as well as land use (LU; filled circles, near-natural habitats; open circles, anthropogenic habitats) along the elevational gradient of Mount Kilimanjaro, Tanzania. The Bayesian hierarchical structural equation models in **b**, **c** tested for direct and indirect 'functional diversity'-mediated effects of mean annual temperature, mean annual precipitation and land use on (**b**) niche breadth (partner diversity, $e^H$) and (**c**) niche partitioning (complementary specialisation, $d'$) of plants and animals via functional diversity of plant and animal communities (functional dispersion, FD; subscripts p and a for plants and animals, respectively). The lines in **a** represent loess smooth functions (degree = 2, span = 0.5) fitted to the temperature and precipitation data across the elevational gradient. In **b**, **c** only paths that were supported by the Bayesian variable selection (2log$_e$(Bayes factor) > 2) are shown (see Supplementary Table 2). Path colours depict bottom-up-mediated effects (blue), top-down-mediated effects (red) and direct abiotic effects (grey) on network structure. Grey double-headed arrows depict covariance terms that account for correlated errors due to common unmeasured sources of variance and due to reciprocal effects of functional diversity on the other trophic level. Path widths are proportional to standardised effect sizes. The values near the endogenous variables depict the marginal ($r^2$) variance explained by fixed factors only, as well as the conditional ($r_c^2$) variance explained by fixed and random factors combined (see Methods for details)[70]. Sample sizes are $n_{obs} = 126$ observations, $n_{site} = 53$ study sites and $n_{mutualism} = 3$ mutualisms

simultaneously bottom-up and top-down controlled, as the functional diversity of plants and animals limited the niche breadth and partitioning of the respective other trophic level. The fact that the niche breadth and partitioning of plants, as well as niche breadth of animals were related to the functional diversity of matching, energy and foraging traits, suggests that multiple assembly processes determine these niche properties[37]. In contrast, the niche partitioning of animals was mainly related to the functional diversity of foraging strata in plant communities (i.e., variability in plant height). This finding supports the idea that vertical stratification in tropical forest ecosystems fosters niche partitioning among different functional guilds in mutualistic networks[16]. Overall, these results question the general prevalence of bottom-up effects of producer diversity on consumers, as suggested by biodiversity experiments[12]. Our results rather suggest that the prevalence of bottom-up and top-down control depends on the abiotic context[10]. Thus, our study demonstrates the value of trait-based network approaches for gaining mechanistic insights into how abiotic constraints on community assembly affect biotic interactions in real-world ecosystems[15].

We found that the functional diversity of plant communities increased with precipitation, while the functional diversity of animal communities increased with temperature. Both the dependence of plant diversity on water availability and the dependence of animal diversity on ambient temperature are well documented patterns in the literature[38–44]. In our study, changes in plant functional diversity were mainly related to an increase in the variability of plant height with precipitation, which may indicate that abiotic filtering in arid ecosystems constrains the range of plant growth forms, whereas in wet ecosystems increased competition (e.g., for light) leads to vertical stratification in plant communities[37,45]. On the other hand, changes in animal functional diversity were primarily related to an increase in the variability of matching traits with temperature, which may indicate that energetic constraints in cold ecosystems favour less selective foraging strategies and less specialised morphologies of

plant–animal mutualisms. In addition, associations between traits related to energy requirements and provisioning could be more strongly affected by the environmental context, if high competition at low resource availability forces animals to forage less selectively[33,34]. The more variable associations of traits related to foraging stratum and animal mobility with network structure, may be due to the fact that animals frequently cross habitat boundaries to obtain resources[35], even though certain bird[16,24] and insect[18–23] species preferentially forage in specific habitats or vegetation layers for flowers and fruits. These results suggest that traits related to size matching between plants and animals, as well as traits related to resource provisioning and energy requirements, have the strongest effect on the assembly of pair-wise interactions and that a few trait dimensions may be sufficient to characterise the functional structure of mutualistic networks[36].

In line with our prediction, changes in the functional diversity of plant and animal communities were the principal drivers of changes in network structure along the studied elevational gradient. More specifically, we discovered that biotic interactions are

traits that are related to resource use[46]. In general, the relationships between plant and animal functional diversity and climatic factors suggest that the niche space of these communities expands and the abundance of functionally distinct species increases under favourable climatic conditions that allow for a wider range of functional strategies[37,45–47], which is consistent with the 'physiological tolerance hypothesis' in plants[39] and the 'energy constraints hypothesis' in animals[46].

The fact that plant and animal functional diversity were related to distinct environmental factors indicates that the responses of both trophic levels to changes in abiotic conditions could be decoupled. Despite these potentially decoupled responses of plants and animals to abiotic changes, the functional diversity of each trophic level reciprocally constrained interactions with the other trophic level via bottom-up and top-down control. According to our findings, the extent to which biotic interactions are bottom-up- or top-down-controlled depends on whether abiotic factors primarily limit plant *or* animal functional diversity. In particular, our results suggest that top-down effects are more limiting in cold and wet ecosystems (at mid and high elevations), whereas bottom-up effects limit network assembly in hot and arid ecosystems (at the mountain base). Interestingly, this pattern resembles the latitudinal shift in the relative importance of precipitation and temperature constraints on plant and animal diversity[38]: While water availability is the principal factor limiting plant and animal diversity in warm tropical and subtropical ecosystems, temperature is more limiting in cold temperate and boreal ecosystems. This suggests that the latitudinal shift in the importance of these abiotic factors for plant and animal diversity might be partly mediated by the relative importance of bottom-up and top-down control in different environments. This context-dependence of the prevailing mechanism is also likely to be an important driver of the observed variation in the effects of global change on biotic interactions and associated ecosystem functions[48].

In addition to the indirect effect of temperature through animal functional diversity, temperature was also directly negatively associated with niche breadth and overlap in the networks. This direct temperature effect may be explained by changes in resource availability and/or foraging behaviour along the temperature gradient[33,34,40,49]. Lower resource availability and increased competition, as well as higher energetic costs of foraging in cold environments, may force animals to forage less selectively on plants and might cause an increase in niche breadth[33,49] and overlap[34] at low temperatures.

We found no consistent effects of land use on multivariate functional diversity of plants or animals across the three mutualisms. However, we detected an increase in functional diversity of traits related to energy requirements of animals in anthropogenic compared with near-natural habitats. Previous studies reported a higher availability of plant resources (flowers and fruits) in anthropogenic compared with near-natural habitats on Mount Kilimanjaro[45,50]. Increased resource availability in anthropogenic habitats may release animal communities from energetic constraints and might allow for a wider range of metabolic niches[51] and an increased variability in animal body sizes[46,50]. Plant functional diversity may be less related to effects of land use, because many of the anthropogenic habitats on Mount Kilimanjaro host a high plant diversity (e.g. the traditional homegarden agroforestry systems in the lower montane forest belt)[52,53].

Here, we integrated multiple types of mutualistic interaction networks with functional traits to assess the effects of trait matching and functional diversity on the assembly of species-rich plant–animal networks. Our study demonstrates that trait matching is a key determinant of network assembly and that the

relative importance of bottom-up and top-down control in mutualistic networks is determined by whether environmental conditions limit the functional diversity of resources or consumers. This has important implications for the response of interaction networks and associated ecosystem functions to environmental change. As species have to adapt to their environment, but also depend on interactions with other species, trait matching and the functional diversity of interaction partners constrain species' responses to environmental change. Trade-offs between functional adaptations to environmental conditions and biotic interactions may cause non-equilibrium dynamics, in which species are far from their optimum in terms of environmental conditions or interaction partners[54]. This trade-off between abiotic and biotic constraints might limit the adaptive capacity of species to environmental change. Therefore, projections of biodiversity and ecosystem functions in response to abiotic changes are inaccurate if they do not account for the manifold interactions among species in ecological communities.

## Methods

**Study area.** The study was conducted on the southern and south-eastern slopes of Mt. Kilimanjaro (Tanzania, East Africa; 2°45′–3°25′S, 37°00′–37°43′E). Mt. Kilimanjaro rises from the savannah plains at an elevation of 700 m a.s.l. to a snow-capped summit at an elevation of 5895 m a.s.l. Precipitation is bimodal with the main rainy season occurring from March through May and more variable short rains around November. The mean annual temperature decreases almost linearly with elevation, whereas mean annual precipitation peaks at an elevation of ~2200 m a.s.l. (Fig. 3a)[52,53]. Due to a long history of human settlement, natural habitats in the lowlands have been subject to various forms of human disturbance including fire, logging, and agroforestry practices[52,53]. Habitats above 2700 m a.s.l. are protected as a national park since 1973 (Mt. Kilimanjaro National Park); since 2006 also the forests above 1800 m a.s.l. are included in the National Park.

**Study design.** We collected data on a total of 53 study sites along five transects on the southern slopes of the mountain (minimum pair-wise distance of 300 m). The study sites cover six near-natural and six anthropogenic habitat types: savanna ($n = 5$) and maize fields ($n = 4$; 870–1150 m a.s.l.); lower montane forest ($n = 5$), traditional homegarden agroforestry systems ($n = 5$), coffee plantations ($n = 5$) and grassland ($n = 7$; 1150–2050 m a.s.l.); natural ($n = 5$) and disturbed *Ocotea* forest ($n = 4$; 2150–2750 m a.s.l.); natural ($n = 5$) and disturbed *Podocarpus* forest ($n = 3$; 2750–3000 m a.s.l.); *Erica* forest ($n = 2$; 3950–4000 m a.s.l.), as well as alpine *Helichrysum* vegetation ($n = 3$; 3850–4400 m a.s.l.). A detailed description of vegetation and land-use types on Mount Kilimanjaro is given by Hemp[52,53]. No statistical methods were used to predetermine sample size.

**Temperature and precipitation data.** All study sites were equipped with temperature sensors that were installed ~2 m above the ground. Temperature sensors measured temperatures in 5 min intervals for a time period of ~2 years. We calculated the mean annual temperature (MAT, °C) as the average of all measurements per study site. Mean annual precipitation (MAP, mm yr$^{-1}$) was interpolated across the study area using a co-kriging approach based on a 15-year data set from a network of about 70 rain gauges on Mt. Kilimanjaro[53]. As we did not have data of MAT and MAP for one study site, we predicted these data using a linear model with the observed MAT and MAP data as the response variables and elevation (third-order polynomial) and habitat type as additive explanatory variables (MAT: $R^2 = 0.99$, $n = 52$, $P < 0.0001$; MAP: $R^2 = 0.98$, $n = 52$, $P < 0.0001$; Fig. 3a).

**Plant–animal interactions.** We studied bird–fruit and bird–flower interactions on 52 of the 53 study sites between November 2013 and October 2015. To do so, we established one plot of 30 × 100 m$^2$ size on each site, covering a representative amount of the flowering and fruiting plant community typical for each habitat type. Each site was sampled once, but replicate sites in each habitat type were sampled both in the cold and in the warm dry season to account for seasonal variability[55]. We observed birds using binoculars to record interactions with fruiting and flowering plants. We identified birds using Zimmerman et al.[56]. On each site, birds were observed for 25 h in total, distributed over 4 consecutive days[55]. Observations were conducted for 7 h (1–5 h after sunrise, 2 h before sunset) on the first 3 days and for 4 h on the last day (1–4 h after sunrise). We recorded the number of visits of each bird species on each fruiting or flowering plant species, respectively, and recorded their behaviour. In the analysis we considered only visits that were classified as legitimate seed dispersal or pollination events, i.e., swallowing or carrying away of fruits from mother plants, as well as pollen uptake. In total, we conducted 1300 h of bird observations during the study period, during which we recorded 9194 bird–fruit interactions between 68 plant and 86 bird species, as well as 3124 bird–flower interactions between 30 plant and 28 bird

species, respectively. We had to restrict our analysis to a subset of 8085 interactions between 63 plant and 85 bird species on 39 study sites in the bird–fruit mutualism and to a subset of 2583 interactions between 26 plant and 24 bird species on 20 study sites in the bird–flower mutualism for which we were able to obtain trait data.

We studied insect–flower interactions on 19 of the 53 study sites between January 2011 and November 2012. To do so, we established one plot of $100 \times 100$ $m^2$ size on each site, covering a representative amount of the flowering plant community typical for each habitat type. If possible, we sampled each site several times to account for seasonal variability between the cold and warm dry seasons. In the analysis, we accounted for the repeated sampling on the study sites by including study site as a random factor (see section Bayesian hierarchical structural equation model). During each sampling round on a given study site, we conducted a 4-h transect walk. In case of rain, strong wind or dense fog, transect walks were interrupted and continued later on that day or on the next day. During each transect walk, we moved slowly through the vegetation of each study site and recorded each interaction in which an insect touched the reproductive part of a plant species (herbaceous plants and bushes up to 9 m height). Thus, we assumed that all flower-visiting insects contribute to pollination. We collected specimens of each plant species for identification in the lab. Whenever possible, flower-visiting insects were caught with sweep nets. Thereby, we restricted our sampling to major groups of pollinators (including Apiformes, the paraphyletic group of non-bee aculeates, Symphyta and Syrphidae). We excluded other Diptera from the analyses, for which identification below family level was not feasible. We identified the caught insect specimens to the lowest taxonomic level possible. We identified 79% of all flower-visiting insects to species level and 97% to genus level (for simplicity, we refer to all morphospecies as species). In total, we conducted 320 h of insect observations during the study period, during which we recorded 4236 insect–flower interactions between 149 plant and 188 insect species. We had to restrict our analysis to a subset of 4060 interactions between 131 plant and 183 insect species on 19 study sites in the insect–flower mutualism for which we were able to obtain trait data.

**Plant and animal functional traits.** We quantified functional traits of plant and animal species that are known to structure the mutualistic interactions between these species groups via trait matching[16–24]. Thereby, we distinguished between three types of traits: traits related to size matching and the effectiveness of interactions (matching traits), traits related to resource provisioning by plants and energy requirements of animals (energy traits), and traits related to foraging stratum and animal mobility (foraging traits). Traits related to size matching were fruit diameter and bill width in the bird–fruit mutualism, corolla depth and bill length in the bird–flower mutualism, and corolla depth and proboscis length in the insect–flower mutualism[17,24]. Traits related to resource provisioning and energy requirements were crop mass and body mass in the bird–fruit mutualism, the number of flowers per plant and body mass in the bird–flower mutualism, and the number of flowers per inflorescence and head width in the insect–flower mutualism. Traits related to foraging stratum and mobility were plant height and the Kipp's index[57] (i.e., the ratio between Kipp's distance and wing length as a measure of the pointedness of the wing) in the bird–fruit and bird–flower mutualisms, as well as plant height and the forewing index (i.e., the ratio of forewing length to body length) in the insect–flower mutualism[21–23]. We selected head width instead of body length as a proxy for energy requirements in the insect–flower mutualism in order to avoid using the same morphological variable two times in the analysis. As body length and head width were highly correlated (Pearson's $r$ between $\log_e$-transformed variables: $r = 0.84$, $n = 91$ species), the decision of whether to include body length or head width did not affect our conclusions.

For fruiting plants, we measured the maximal diameter of 15 fruits for each plant species (five fruits each from three different individuals) using sliding callipers (precision of 0.01 mm) and weighed the fruits using a digital scale (precision 0.01 g). Moreover, we estimated the total number of ripe fruits per plant individual for each species on the study sites. On plants with very large crop sizes, we counted the number of fruits for representative branches and used these to estimate the crop size of the whole plant. We calculated crop mass by multiplying the mean fruit mass by the mean number of fruits per individual for each plant species. We measured plant height for each species on the study sites using a laser range finder (precision 1 m).

For flowering plants, we measured corolla depth using sliding callipers (precision 0.01 mm) on herbarium specimens. For the bird–flower mutualism, we estimated the total number of flowers per plant individual for each species on the study sites and measured plant height using a laser range finder (precision 1 m). For the insect–flower mutualism, we counted the number of flowers per inflorescence and measured plant height on herbarium specimens or compiled data about plant height from the literature if only parts of plant specimen were available in herbarium samples[58]. When no trait information was available for a plant species, we used the trait values from closely related species in the same genus (bird–flower mutualism: $n = 4$ out of 31 plant species; insect–flower mutualism: $n = 7$ out of 136 plant species).

For fruit-eating and flower-visiting birds, we measured bill width, bill length, Kipp's distance and wing length using sliding callipers (precision 0.01 mm) on museum specimens following Eck et al.[57] and extracted information about body

mass from the literature[59,60]. Measurements were taken on an average of four specimens per species (range = 1–16). We measured bill length as the distance from the commissural point of the upper and lower bill to the tip of the closed bill, and bill width as the external distance between the two commissural points. We calculated Kipp's index as the ratio of Kipp's distance (distance between tip of the first secondary and tip of the longest primary of the folded wing) and wing length.

For flower-visiting insects, we measured proboscis length (Hymenoptera: length of glossa; Diptera: length of labellum)[61], forewing length, body length, head width and intertegular distance (for aculeate bees) of collected specimens using a binocular microscope with a calibrated ocular micrometre (precision 0.01 mm). When no trait information was available for an insect species, we used the mean trait values from related species in the same genus ($n = 11$ cases), from species in the same family ($n = 12$ cases), or from species in the same order ($n = 2$ cases). Excluding the insect species for which we only had trait information at the genus, family or order level from the analyses led to identical conclusions. For 75 bee species, we estimated body length based on the relationship with intertegular distance ($\log_e$(body length) $= 1.89 + 0.518 \times \log_e$(intertegular distance), $r^2 = 0.48$, $n = 49$, $P < 0.0001$). For 10 hymenoptera species (families: Pompilidae, $n = 3$; Tiphiidae, $n = 2$; and Vespidae, $n = 5$), we estimated proboscis length based on the relationship with head width ($\log_e$(proboscis length) $= -1.32 + 1.54 \times \log_e$(head width), $r^2 = 0.47$, $n = 112$, $P < 0.0001$). We calculated the forewing index as the ratio of forewing length and body length.

We square-root-transformed all plant and animal traits for the bird–fruit, the bird–flower and the insect–flower mutualisms before the analysis.

**Trait associations in mutualistic networks.** To test for functional relationships between the multidimensional trait spaces of plants and animals in the mutualistic networks, we adopted a combination of the RLQ and the fourth-corner analyses[24–26]. The RLQ analysis builds on the simultaneous ordination of three tables: a table $\mathbf{R}$ ($m \times p$) describing $p$ traits for $m$ plant species, a table $\mathbf{Q}$ ($n \times a$) describing $a$ traits for $n$ animal species, and a third table $\mathbf{L}$ ($m \times n$) containing qualitative or quantitative information about the occurrence or frequency of pairwise interactions between the $m$ plant and $n$ animal species. Here we defined $\mathbf{L}$ as a binary matrix based on whether an interaction between a pair of plant and animal species had been observed at least once across all study sites in each mutualism. Therefore, table $\mathbf{L}$ is analogous to a metaweb that describes the potential for pairwise interactions between plants and animals based on all available information from the observations. We first applied correspondence analysis (CA) to table $\mathbf{L}$ and principal components analyses (PCA) to tables $\mathbf{R}$ and $\mathbf{Q}$[25]. In the PCAs of $\mathbf{R}$ and $\mathbf{Q}$ each plant and animal species was weighted by its number of links in $\mathbf{L}$ (i.e., species degree)[62]. Then, we combined the three separate ordinations of $\mathbf{R}$, $\mathbf{L}$, and $\mathbf{Q}$ using the RLQ approach to identify the main relationships between plant and animal trait spaces as mediated by their interactions[25]. The RLQ analysis is based on a $p \times a$ matrix $\boldsymbol{\Omega}$ containing measures of the intensity of the links between plant and animal traits[63]. The further eigendecomposition of $\boldsymbol{\Omega}^T \boldsymbol{\Omega}$ allows identifying the main associations between plant and animal traits[26,64]. For the first dimension, this analysis finds a vector $\mathbf{u}_1$ containing coefficients for the plant traits and a vector $\mathbf{v}_1$ of coefficients for the animal traits. These loadings measure the contributions of individual traits and are used to compute scores for plant ($\mathbf{x}_1 = \mathbf{RD}_p\mathbf{u}_1$) and animal species ($\mathbf{y}_1 = \mathbf{QD}_a\mathbf{v}_1$) where $\mathbf{D}_p$ and $\mathbf{D}_a$ are diagonal matrices of variable weights (here species degree). RLQ chooses the coefficient vectors $\mathbf{u}_1$ and $\mathbf{v}_1$ so that the derived species scores have maximum squared cross-covariance $\text{cov}_\mathbf{P}(\mathbf{x}_1, \mathbf{y}_1)^2 = (\mathbf{x}_1^T \mathbf{Py}_1)^2 = \lambda_1$, where $\lambda_1$ is the first RLQ eigenvalue. In other words, RLQ finds linear combinations of plant and animal traits (i.e., trait syndromes) so that their squared cross-covariance is maximum. The same quantity is maximised for the $k$ dimensions with the additional constraints of orthogonality ($\mathbf{u}_i^T \mathbf{D}_p\mathbf{u}_j = \mathbf{v}_i^T \mathbf{D}_a\mathbf{v}_j = 0$ for $i \neq j$). Results are stored in matrices $\mathbf{U} = [\mathbf{u}_1 |...| \mathbf{u}_k]$, $\mathbf{V} = [\mathbf{v}_1 |...| \mathbf{v}_k]$, $\mathbf{X} = \mathbf{RD}_p\mathbf{U} = [\mathbf{x}_1 |...| \mathbf{x}_k]$ and $\mathbf{Y} = \mathbf{QD}_a\mathbf{V} = [\mathbf{y}_1 |...| \mathbf{y}_k]$.

We used the eigenvalues of the RLQ analysis to select those ordination axes that explained most of the cross-covariance between plant and animal trait spaces for our analyses[25]. We selected the first two RLQ axes, because in all three mutualisms these axes together explained more than 99% of the cross-covariance between plant and animal trait spaces (bird–fruit: 90% and 9.3%; bird–flower: 99% and 0.75%; insect–flower: 94.8% and 5.1%, Fig. 2a, c, e). Then, we applied the fourth-corner permutation test (model 6 with 9999 permutations)[25] to evaluate the statistical significance of the associations between plant and animal trait spaces using three different approaches. First, we used the sum of eigenvalues of the RLQ analysis as a multivariate statistic to test for global associations among plant and animal trait spaces[25,26]. Second, we tested for correlations between the first and second dimensions of plant and animal trait spaces by using the RLQ scores on the first two axes of the plant and animal trait spaces as variables in the fourth-corner analysis[25]. Third, we assessed which of the different trait types (matching, energy and foraging traits) were correlated with each of the first two RLQ axes[25]. Because we aimed at generalising our results across mutualisms and because the direction of effects in multivariate space is arbitrary, we compared the absolute magnitude of correlations between matching, energy and foraging traits and the RLQ axes across the three mutualisms. Moreover, instead of assessing the significance of individual correlations, we assessed the overall support for the hypotheses that matching, energy and foraging traits are related to the first and second RLQ axes across the three mutualisms. To do so, we used the equation given by Moran[28], based on a Bernoulli process, to calculate the probability, $P$, of obtaining a given number of significant tests from a given

number of trials just by chance. This probability is given by the equation $P = [N! / (N - K)! K!] \times \alpha^{K}(1 - \alpha)^{N-K}$, where $N$ is the number of tests conducted and $K$ is the number of tests below the significance level $\alpha$ (here $\alpha = 0.05$). The rationale behind this equation is that the evidence against the null hypothesis from a given number of statistical tests increases with the number of significant tests[28]. For instance, four out of six matching traits were significantly correlated with the first RLQ axes at $\alpha = 0.05$ (Supplementary Table 1). According to Moran's equation the probability of obtaining four significant results out of six tests at $\alpha = 0.05$ by chance is $P = 8.5 \times 10^{-5}$, providing strong support for the hypothesis that matching traits structure pair-wise interactions in mutualistic networks.

Previous work has shown that the sequential permutation approach for statistical testing of the 'fourth-corner problem' (model 6) has good power (0.88) for 100 species, reasonable power (0.60) for 50 species and some power (0.40) for 30 species[27]. As the number of plant and animal species in two of our datasets is relatively small, the statistical power of the permutation test is low. For instance, although the magnitude of the absolute correlations between matching traits and RLQ axis 1 in the bird–flower mutualism is similar to those in the bird–fruit and insect–flower mutualisms, the permutation test is not significant at $\alpha = 0.05$ (Supplementary Table 1; Supplementary Fig. 1). Therefore, the statistical tests we conducted can be considered highly conservative with respect to Type I errors.

**Functional diversity effects on network assembly**. We used functional dispersion to measure the functional diversity of plants and animals within the mutualistic networks (FD$_p$ and FD$_a$ hereafter)[65]. Compared to other measures of functional diversity functional dispersion has several advantages: Functional dispersion is abundance-weighted and therefore less influenced by extreme values and it is by definition unaffected by species richness[65]. We calculated multivariate and univariate functional dispersion of plant and animal communities in each mutualism. To do so, we first calculated distance matrices based on the Gower distance between species based on the combination of all three functional traits (multivariate FD) or based on each trait type separately (univariate FD based on matching, energy or foraging traits). The Gower distance equals the mean character difference across traits after standardisation of the trait values by their ranges and has been recommended for calculation of functional diversity metrics based on multiple traits, because it is less sensitive to extreme trait values than the Euclidean distance[66]. Moreover, the standardisation of the trait values by their ranges yields an empirical maximum value of the distance function that equals one[66], which allows for a meaningful comparison among multiple species groups with different sets of functional traits. Then we projected species into a multidimensional functional trait space using a principal coordinates analysis (PCoA), and calculated functional dispersion as the mean abundance-weighted distance of each species in a given community to the abundance-weighted centroid of all species in this community[65]. To estimate the abundance of plant and animal species in the networks, we used their marginal interaction totals in each network. Functional dispersion was only weakly correlated with species richness of plants and animals in our data (absolute Pearson's $|r| < 0.41$ for plants and $|r| < 0.33$ for animals across multivariate and univariate FD metrics; $n = 126$ in all cases).

To quantify the interaction niches of plants and animals in the networks, we used two different measures of niche breadth and niche complementarity. We quantified niche breadth of plants and animals as the mean effective number of partners based on the exponent of the Shannon diversity of links ($e^{H_p}$ and $e^{H_a}$), and niche partitioning as the mean standardised Kullback–Leibler distance ($d'_p$ and $d'_a$) across the plants and animals in each network, respectively[29].

**Bayesian hierarchical structural equation model**. To test for bottom-up and top-down effects of plant and animal functional diversity on niche breadth and partitioning, respectively, we used Bayesian hierarchical structural equation models. The general structure of the models that we used here is reviewed in the literature[67] and described in more detail in the supplementary material (Supplementary Note 1).

We fitted two separate structural equation models, one including $e^{H_p}$ and $e^{H_a}$ and the other including $d'_p$ and $d'_a$ as measures of the interaction niches of plant and animal communities on each study site (Supplementary Fig. 2). In these structural equation models, we treated mean annual temperature (MAT, °C), mean annual precipitation (MAP, mm yr$^{-1}$) and land use (LU, binary variable) as exogenous predictor variables. We treated FD$_p$ and FD$_a$ as well as measures of plant and animal interaction niches ($e^{H_p}$ and $e^{H_a}$ or $d'_p$ and $d'_a$, respectively) as endogenous variables. The models included all potential direct effects of MAT, MAP and LU on FD$_p$ and FD$_a$, as well as on $e^{H_p}$ and $e^{H_a}$ or $d'_p$ and $d'_a$, respectively. Moreover, the models included the effects of FD$_p$ and FD$_a$ on $e^{H_p}$ and $e^{H_a}$ or $d'_p$ and $d'_a$, respectively. We also included covariance terms between FD$_p$ and FD$_a$, as well as between $e^{H_p}$ and $e^{H_a}$ or $d'_p$ and $d'_a$ to account for correlated errors due to common unmeasured sources of variance and due to reciprocal effects of functional diversity on the other trophic level. The total number of samples included in the analysis was $n = 126$. To account for the hierarchical structure of the data we included study site ($n = 53$) and mutualism type ($n = 3$) as random factors into the structural equations. The measures $e^{H_p}$ and $e^{H_a}$ were transformed to their natural logarithm and all variables were scaled to zero mean and unit variance before analysis.

To separate informative from non-informative paths, we used a Bayesian indicator variable selection with global adaptation[68] (Supplementary Note 1). We

used 2log$_e$(Bayes factor) as a measure of evidence for a given effect (BF hereafter)[69]. Values of BF < 2 indicate no support; values between 2 and 6 indicate positive support; values between 6 and 10 indicate strong support; and values >10 indicate decisive support. We report the marginal variance, $r_m^2$, that is explained by the fixed factors, as well as the conditional variance, $r_c^2$, that is explained by the fixed and random factors combined as measures of model fit[70].

The models were implemented in JAGS[71], and run in R[72] through the rjags package[73]. We ran eight parallel chains for the models. We used uninformative priors for all parameters and the initial values for the chains were drawn randomly from uniform distributions. Each chain was run for 51,000 iterations with an adaptive burn-in phase of 1000 iterations and a thinning interval of 100 iterations, resulting in 500 samples per chain, corresponding to 4000 samples from the posterior distribution. The chains were checked for convergence, temporal autocorrelation, and effective sample size using the coda package[74]. Residuals were checked for normality and variance homogeneity.

**Code availability**. The computer code of the analyses is available in figshare with the identifier https://doi.org/10.6084/m9.figshare.6633032. The JAGS code for the Bayesian hierarchical structural equation model is also given as part of the Supplementary Information (Supplementary Note 1).

**Data availability**. The data that support the findings of this study are available in figshare with the identifier https://doi.org/10.6084/m9.figshare.6633032. Due to the Data and Publication Policy of the Research Unit FOR1246, the figshare data are embargoed for public release until 1 January 2020. Until the embargo date for public release the data are available from the corresponding author upon reasonable request.

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

## Acknowledgements

We thank the Tanzanian Commission for Science and Technology, the Tanzania Wildlife Research Institute and the Kilimanjaro National Park authority for their support and for granting us access to the Kilimanjaro National Park area. We are grateful to all companies and private farmers that allowed us to work on their land. We thank the KiLi field staff for helping with data collection at Mt. Kilimanjaro. This study was conducted within the framework of the Research Unit FOR1246 (Kilimanjaro ecosystems under global change: linking biodiversity, biotic interactions and biogeochemical ecosystem processes,

https://www.kilimanjaro.biozentrum.uni-wuerzburg.de) funded by the German Research Foundation (DFG).

## Author contributions

J.A., K.B.G., and M.S. conceived the study. M.F., A.H., and I.S.D. initiated the research unit at Mt. Kilimanjaro. T.N. and A.H. collected climate data. A.C. and M.G.R.V. collected interaction data. J.A., A.C., A.M., N.M., D.S.C., and M.G.R.V. measured plant and animal traits. J.A. processed and analysed the data and wrote the first version of the manuscript with input from M.S. H.I.D., K.M.H., M.K., M.K.P., M.T. and all authors contributed to the final version of the manuscript.

## Additional information

**Competing interests:** The authors declare no competing interests.

