## [Peer Review File · Nature Communications]

Reviewers' comments:

Reviewer #1 (Remarks to the Author):

In this study, the authors studied the pairwise characteristics of mutualist networks along an elevation gradient. The data included bird-fruit, bird-flower, and insect-flower mutualisms (sensu lato). Hypotheses about network relations were evaluated using multivariate techniques. A specific hypothesis about top-down versus bottom-up influences on functional diversity and then indirectly on niche breadth and complementarity was specified and evaluated. This latter analysis ultimately provided a very nice and clear summary of the primary and contrasting roles of temperature and precipitation on top-down versus bottom-up influences (Fig. 3).

The power of this study lies in its efforts to integrate ecological networks with a trait-based understanding of mutualisms with a synthetic view of top-down versus bottom-up controls. Ecologists are, of course, interested in understanding ecological systems, but it is another matter to have the data and methodological tools and skills to provide synthetic, quantitative views. The authors are to be commended for helping to move our understanding to this “next level”. The findings are nicely complex, but reasonably simple at the same time. This is accomplished through careful choices about model architecture. Nicely done!

I find the paper to be well thought out and well presented. I find no problems with the analyses or inferences drawn from them. All my comments are minor.

- (1) Title is weak. I believe a more compelling statement of your findings is needed. Right now it just tells us that you found reciprocal effects in mutualisms, which is the definition of mutualisms.
- (2) Figure 1 needs to be redrawn so that animals are on top and plants on bottom. The top-down and bottom up conceptual structure is used everywhere else in the paper except in this one figure. Compare to Figure 3, for example.
- (3) A very key part of this paper is the terminal variable pair, e^H and d^H . The one place in the paper where anything of substance is said about these variables, lines 392-396 in the Methods section, is insufficient. Too much of a burden is placed on the reader to already know how to interpret the proposed impacts of functional diversity on partner e^H and d^H . There should be a further description of how to interpret these responses in the main text, but also a bit more help in the caption of Figure 3.
- (4) Line 145 – Pet peeve of mine is when authors try to justify the importance of their study by claiming to be “the first to . . .”. This absolute statement (a) implies you have read all the literature, which is impossible, (b) creates some impossible standard of comparison because all studies are unique at some level and non-unique at another level of description, and (c) obstructs a clear statement of how your study advances our understanding.
- (5) Line 359 – Move “most” to follow “the”.

(6) I did not find a statement of data availability. As a reviewer I will be asked if the data are made available.

Ultimately, I think this is a great study and a great paper. Impressive!

Reviewer #2 (Remarks to the Author):

The paper proposes a nice theoretical ecological framework that is tested on a fantastic data set. It aims to relate traits and functional diversity to the structure of mutualistic networks (interaction niches).

I found the paper well written and the theoretical framework really nice. Concerning the statistical analysis (which is my real field of expertise), I found the paper quite clear and the choice of methods is appropriate. I just have a minor comment concerning the RLQ/fourth-corner analysis. The authors tested the significance of axes correlations by setting the argument `typetest="axes"` in the `ade4` function. In fact, this test is not correct (it is only produce for pedagogical purpose, I will change the code of the function to avoid such misuse). They should use "Q.axes" and "R.axes" to test the effect of each types of traits on the global link. I would assume, that no traits would be linked to the second axis as all the structure seems summarize on only one dimension.

S. Dray

Reviewer #3 (Remarks to the Author):

In this paper the authors seek to understand to what extent shifts in plant and animal functional diversity along environmental gradients alters the structure of species interaction networks and whether network assembly is bottom-up, top-down, or both. Few studies have simultaneously examined the contribution of bottom -up and top-down processes and this study here provides a novel examination of this. Moreover, this is an important question that has not been addressed from a functional perspective. The authors have an excellent dataset for addressing this question and robustly test these ideas. Overall, I think this is an interesting study, but there are several conceptual issues that I believe need to be addressed.

The biggest issues I see here is that not quite enough space is used to explain the conceptual framework proposed in Figure 1. I understand that space constrains, but some things need to be clarified for a general audience. First the term partner diversity is not defined here. It has a specific definition and for those that are not familiar with the field, it needs to be explained. Next, Fig. 1d refers to functional diversity of trophic level A and B, Figs 1A-C have A for animal and P for plant. This could lead to confusion ad people will think Level A refers to animal functional diversity. Next, the proposed figure also has implications for horizontal interactions. In Fig1B the loss of P3 leads to increased niche overlap among A1 and A2 and, in theory, increasing competition which should influence the functional diversity patterns of the Animal community. So, the loss of these mutualisms may lead to increased competition,

yet competition is barely mentioned here and the assumption is that only abiotic factors influence the functional diversity of the horizontal communities when we know competition and facilitation do as well. My feeling is that there is a bit more complexity here that needs to be explained. I think the authors do a pretty good job given the space constraints, but a bit more is needed.

My next concern is that more details are needed about the functional diversity patterns. Some studies have found that functional diversity patterns are only revealed when examining multivariate trait patterns (Kraft et al. 2015 PNAS). Other studies have found that multivariate trait patterns mask functional diversity patterns due to opposing responses of different traits (Spasojevic and Suding 2012 Journal of Ecology). Without seeing any of the individual trait patterns it is hard to assess which is the case. Might some of the lower r^2 values be caused by trait patterns masking each other? Are some traits more or less important? How are these individual traits changing along the elevation gradient? Could land use be having opposing effects on different traits resulting in a non-significant effect? This latter possibility seems to be suggested by the authors on lines 216-218. The individual trait patterns, in my opinion, are much more important supplementary information than supplementary figure 2. Functional dispersion is known to be independent of species richness, so it is expected that there is no relationship. This figure is not needed.

Overall the analysis is well done and comprehensive. The only question I have is if non-linear relationships were explored for the paths in the SEM – particularly for precipitation?

In conclusion, I think this is an interesting and important study that would benefit from a few clarifications.

Minor suggestions

Line 35: change “which” extent to “what” extent

Line 38: add “their” before associated

Figure 3. In the text please define the differences between rm^2 and rc^2 .

We thank the Reviewers for their constructive and positive comments, which in our opinion helped to improve the manuscript. Below we provide a point-by-point response to the comments, which are highlighted in red font. To facilitate the review process, we have also highlighted the changes made to the manuscript in red font in the Word document. Please note that line numbers in our responses refer to lines in the revised version of the manuscript.

Reviewers' comments:

Reviewer #1 (Remarks to the Author):

In this study, the authors studied the pairwise characteristics of mutualist networks along an elevation gradient. The data included bird-fruit, bird-flower, and insect-flower mutualisms (sensu lato). Hypotheses about network relations were evaluated using multivariate techniques. A specific hypothesis about top-down versus bottom-up influences on functional diversity and then indirectly on niche breadth and complementary was specified and evaluated. This latter analysis ultimately provided a very nice and clear summary of the primary and contrasting roles of temperature and precipitation on top-down versus bottom-up influences (Fig. 3).

The power of this study lies in its efforts to integrate ecological networks with a trait-based understanding of mutualisms with a synthetic view of top-down versus bottom-up controls. Ecologists are, of course, interested in understanding ecological systems, but it is another matter to have the data and methodological tools and skills to provide synthetic, quantitative views. The authors are to be commended for helping to move our understanding to this “next level”. The findings are nicely complex, but reasonably simple at the same time. This is accomplished through careful choices about model architecture. Nicely done!

I find the paper to be well thought out and well presented. I find no problems with the analyses or inferences drawn from them. All my comments are minor.

Response: Thank you for this enthusiastic feedback on our manuscript.

(1) Title is weak. I believe a more compelling statement of your findings is needed. Right now it just tells us that you found reciprocal effects in mutualisms, which is the definition of mutualisms.

Response: We disagree with the reviewer on this point. In our opinion the title well reflects the fact that we are looking at the reciprocal *effects of plant and animal functional diversity* on network assembly. Thus, the title goes beyond merely looking at reciprocal effects in mutualisms, but illustrates that the study aims at providing a trait-based understanding of mutualisms in a community context (as acknowledged by the reviewer). Thus, we would prefer to leave the title unchanged. If the reviewer and the editor insist on changing the title, we would propose '*Abiotic context shapes reciprocal effects of plant and animal functional diversity on mutualistic network assembly*' or '*Abiotic context shapes bottom-up and top-down effects of functional diversity on mutualistic network assembly*' as alternative titles.

(2) Figure 1 needs to be redrawn so that animals are on top and plants on bottom. The top-down and bottom up conceptual structure is used everywhere else in the paper except in this one figure. Compare to Figure 3, for example.

Response: We changed the figure according to your suggestions.

(3) A very key part of this paper is the terminal variable pair, e^H and d^H . The one place in the paper where anything of substance is said about these variables, lines 392-396 in the Methods section, is insufficient. Too much of a burden is placed on the reader to already know how to interpret the proposed impacts of functional diversity on partner e^H and d^H . There should be a further description of how to interpret these responses in the main text, but also a bit more help in the caption of Figure 3.

Response: Thank you for bringing this lack of clarity to our attention (also pointed out by reviewer #3). In response to your comment and the comment of reviewer #3, we added definitions of the terms 'niche breadth' and 'niche partitioning' in the context of species interaction networks at first mentioning in the first paragraph of the introduction (Lines 55–56). Moreover, we added a sentence to the respective part in the results section explaining how we quantified the two properties of the interaction niches (Lines 143–146). For consistency, we exchanged the terms 'partner diversity' and 'complementary specialization' by 'niche breadth' and 'niche partitioning' throughout the text. Moreover, in the figure and table captions we use the terms 'niche breadth and partitioning' followed by the more technical descriptions in brackets 'niche breadth (partner diversity, e^H) and niche partitioning

(complementary specialization, d')' to assure that non-specialist readers can follow the line of argument (see Figs. 1 and 3, Supplementary Fig. 3 and 4, as well as Supplementary Table 2).

(4) Line 145 – Pet peeve of mine is when authors try to justify the importance of their study by claiming to be “the first to . . .”. This absolute statement (a) implies you have read all the literature, which is impossible, (b) creates some impossible standard of comparison because all studies are unique at some level and non-unique at another level of description, and (c) obstructs a clear statement of how your study advances our understanding.

Response: We rephrased the sentence, which now reads „Our study provides a general assessment of...” (Line 171).

(5) Line 359 – Move “most” to follow “the”.

Response: We changed the wording of the sentence. The sentence now reads: “...select those ordination axes that explained most of the cross-covariance...” (Line 414).

(6) I did not find a statement of data availability. As a reviewer I will be asked if the data are made available.

Response: Thank you for bringing this obvious lack of reproducibility to our attention. All data and code supporting the main findings of the paper will be made available in a public repository upon publication of the paper. We added a data and code availability statement to the methods section (Lines 521–523).

Ultimately, I think this is a great study and a great paper. Impressive!

Response: We very much appreciate these encouraging comments on the paper.

Reviewer #2 (Remarks to the Author):

The paper proposes a nice theoretical ecological framework that is tested on a fantastic data set. It aims to relate traits and functional diversity to the structure of mutualistic networks (interaction niches).

Response: We appreciate these positive comments on the manuscript.

I found the paper well written and the theoretical framework really nice. Concerning the statistical analysis (which is my real field of expertise), I found the paper quite clear and the choice of methods is appropriate. I just have a minor comment concerning the RLQ/fourth-corner analysis. The authors tested the significance of axes correlations by setting the argument `typetest="axes"` in the `ade4` function. In fact, this test is not correct (it is only produce for pedagogical purpose, I will change the code of the function to avoid such misuse). They should use "Q.axes" and "R.axes" to test the effect of each types of traits on the global link. I would assume, that no traits would be linked to the second axis as all the structure seems summarize on only one dimension.

Response: Thank you for these insightful comments. We partly agree with your comment and, in response to your suggestion, used the proposed tests to assess the effect of each type of traits on the first and second axes of plant and animal trait spaces (Results: Lines 127–135; Discussion: Lines 179–203; Methods: Lines 424–451; Supplementary Table 1; Supplementary Fig. 1). Indeed, this analysis slightly changed the interpretation of the results, as it turned out that the first RLQ axes were most strongly correlated with matching and energy traits, whereas foraging traits were more weakly correlated with the first RLQ axis. Therefore, your comment greatly improved the first part of the manuscript.

However, we are unsure whether the test that we performed in the original version of the manuscript (argument `typetest = "axes"`) is indeed invalid. If we understood the method correctly, the argument `typetest = "axes"` in the `ade4` function, performs a test of whether the first (or second) axis of table Q is correlated with the first (or second) axis of table R. If this is true, then in our case this corresponds to a test, which assesses the correlation between the individual ordination axes of plant and animal trait spaces. This test, therefore, can be useful to disentangle the individual contributions of each RLQ axis to the global link. Of course, the RLQ eigenvalues provide similar information, but the advantage of assessing the correlations between individual axes using the fourth-corner method and Pearson's correlation coefficient is that we obtain an easily interpretable measure of effect size (Pearson's r), as well as an

associated test statistic. Therefore, in our opinion, the test that we performed is correct, but tries to answer a different question: While the arguments “Q.axes” / “R.axes” test for correlations of traits with the ordination axes of tables Q and R; the argument “axes” tests for correlations between the ordination axes of tables Q and R. We would greatly appreciate if you could provide feedback on our line of argument and would support our interpretation that these tests provide complementary information.

If you agree with our argumentation, then we would like to keep the original test for correlations between the ordination axes of tables Q and R in addition to the test that you have suggested. The logical procedure of the RLQ analysis would then be a three-step approach: 1.) Assess presence of global link (based on sum of RLQ eigenvalues); 2.) Assess contribution of individual ordination axes of tables R and Q to global link (based on ordination scores); and 3.) Assess the correlation of the different types of traits with the ordination axes (based on ordination scores and traits).

Reviewer #3 (Remarks to the Author):

In this paper the authors seek to understand to what extent shifts in plant and animal functional diversity along environmental gradients alters the structure of species interaction networks and whether network assembly is bottom-up, top-down, or both. Few studies have simultaneously examined the contribution of bottom -up and top-down processes and this study here provides a novel examination of this. Moreover, this is an important question that has not been addressed from a functional perspective. The authors have an excellent dataset for addressing this question and robustly test these ideas. Overall, I think this is an interesting study, but there are several conceptual issues that I believe need to be addressed.

Response: We thank the reviewer for this positive feedback.

The biggest issues I see here is that not quite enough space is used to explain the conceptual framework proposed in Figure 1. I understand that space constrains, but some things need to be clarified for a general audience. First the term partner diversity is not defined here. It has a specific definition and for those that are not familiar with the field, it needs to be explained. Next, Fig. 1d refers to functional diversity of trophic level A and B, Figs 1A-C have A for animal and P for plant. This could lead to confusion ad people will think Level A refers to

animal functional diversity. Next, the proposed figure also has implications for horizontal interactions. In Fig1B the loss of P3 leads to increased niche overlap among A1 and A2 and, in theory, increasing competition which should influence the functional diversity patterns of the Animal community. So, the loss of these mutualisms may lead to increased competition, yet competition is barely mentioned here and the assumption is that only abiotic factors influence the functional diversity of the horizontal communities when we know competition and facilitation do as well. My feeling is that there is a bit more complexity here that needs to be explained. I think the authors do a pretty good job given the space constraints, but a bit more is needed.

Response: We highly appreciate these constructive comments. In response to your comment and the comment of reviewer #1, we clarified the relations between niche breadth and partner diversity, as well as between niche partitioning and complementary specialization by adding definitions of the terms in the first paragraph of the introduction (Lines 55–56). Moreover, we added a sentence to the respective part in the results section explaining how we quantified the two properties of the interaction niches (Lines 143–146). For consistency, we exchanged the terms ‘partner diversity’ and ‘complementary specialization’ by ‘niche breadth’ and ‘niche partitioning’ throughout the text. Moreover, in the figure and table captions we use the terms ‘niche breadth and partitioning’ followed by the more technical descriptions in brackets ‘niche breadth (partner diversity, e^H) and niche partitioning (complementary specialization, d')’ to assure that non-specialist readers can follow the line of argument (see Figs. 1 and 3, Supplementary Fig. 3 and 4, as well as Supplementary Table 2).

Following your suggestion, we modified Fig. 1d, in which we now refer to the effects of trophic level X on trophic level Y.

We fully agree with your comment regarding the effect of functional diversity in one trophic level on competitive/facilitative interactions among species in the other trophic level. Following your suggestion, we expanded the second paragraph of the introduction and briefly discuss the implications of a loss of functional diversity in one trophic level on competitive interactions in the other trophic level (Lines 66–70). We also added a short note on the implications of our conceptual framework for competitive interactions within trophic levels to the caption of Figure 1. We believe that your comment helped to improve the depth and relevance of the framework.

My next concern is that more details are needed about the functional diversity patterns. Some studies have found that functional diversity patterns are only revealed when examining multivariate trait patterns (Kraft et al. 2015 PNAS). Other studies have found that multivariate trait patterns mask functional diversity patterns due to opposing responses of different traits (Spasojevic and Suding 2012 Journal of Ecology). Without seeing any of the individual trait patterns it is hard to assess which is the case. Might some of the lower r^2 values be caused by trait patterns masking each other? Are some traits more or less important? How are these individual traits changing along the elevation gradient? Could land use be having opposing effects on different traits resulting in a non-significant effect? This latter possibility seems to be suggested by the authors on lines 216-218. The individual trait patterns, in my opinion, are much more important supplementary information than supplementary figure 2. Functional diversity is known to be independent of species richness, so it is expected that there is no relationship. This figure is not needed.

Response: We thank the reviewer for this constructive comment. As suggested we added structural equation models testing for trait-specific bottom-up and top-down effects of functional diversity on network assembly (Methods: Lines 457–462; see Supplementary Figure 4). We describe the results of this additional analysis at the end of the results section (Lines 157–168), and briefly discuss the implications at specific points in the discussion (Lines 208–215, 225–232 and 265–271). Although this additional analysis did not change any of our initial conclusions, it indeed provided some additional insights and allowed for more detailed interpretations of the results in the discussion (as suggested by the reviewer). Even though the univariate models reveal that environmental conditions affect the functional diversity of specific plant and animal traits, we also note that the r^2 -values in the multivariate analyses are always larger than in the univariate models. This suggests that the effects of the environment on functional diversity and the effects of functional diversity on species' interaction niches are best described in multivariate trait space.

We also followed your suggestion and removed Supplementary Figure 2 and only mention correlation coefficients in the Methods section (Lines 473–475).

Overall the analysis is well done and comprehensive. The only question I have is if non-linear relationships were explored for the paths in the SEM – particularly for precipitation?

Response: We did not test for non-linear relationships, because the absence of curvature in the partial residual plots (see Supplementary Fig. 1) suggested that linear relationships adequately describe the structure in the data. To illustrate this we provide a short R-script, in which we show, based on a simulated dataset, that a miss-specified model could be unambiguously identified using partial residual plots (see R code and Fig.1 at the end of the response letter).

Nevertheless, we conducted an additional analysis, in which we tested for each of the endogenous variables (FD_p , FD_a , e^{Hp} , e^{Ha} , d'_p , d'_a), whether on top of the linear effects of temperature and precipitation quadratic effects for both variables improve model fit. The models indicated that quadratic effects do not significantly improve model fit, which corroborates our conclusion based on the visual inspection of the partial residual plots.

In conclusion, I think this is an interesting and important study that would benefit from a few clarifications.

Minor suggestions

Line 35: change “which” extent to “what” extent

Response: Done.

Line 38: add “their” before associated

Response: As this pronoun is not essential to understand the sentence, we would like to omit it due to the length limitation in the abstract.

Figure 3. In the text please define the differences between rm2 and rc2.

Response: The differences between the two measures were already defined in the last sentence of the caption of figure 3, which the reviewer might not have noticed because of the page break. In the figure caption we briefly describe the difference between the two values and also refer to the methods section, and the original work by Nakagawa and Schielzeth (2013).

Literature cited:

R code:

```
##### simulate a dataset
# linear effect for z ~ y
# linear & quadratic effects for z ~ x + x^2
n <- 100
x <- rnorm(n, 0, 1)
y <- rnorm(n, 0, 1)
z <- x + x^2 + y + rnorm(n, 0, 1)
data1 <- data.frame(x = x, y = y, z = z)
pairs(data1)

##### wrong model specification
# only linear effects for x and y
m1 <- lm(z ~ x + y, data1)
summary(m1)
# calculate partial residuals for plotting
data1$resX1 <- with(data1, z - model.matrix(m1)[, -2] %% coef(m1)[-2])
data1$resY1 <- with(data1, z - model.matrix(m1)[, -3] %% coef(m1)[-3])

##### correct model specification
# linear effect for y
# linear & quadratic effects for x
m2 <- lm(z ~ x + I(x^2) + y, data1)
summary(m2)
# calculate partial residuals for plotting
data1$resX2 <- with(data1, z - model.matrix(m2)[, -(2:3)] %% coef(m2)[-(2:3)])
data1$resY2 <- with(data1, z - model.matrix(m2)[, -4] %% coef(m2)[-4])

par(mfrow = c(2, 2), las = 1, cex = 0.7,
    mar = c(4, 4, 1, 1) + 0.1, mgp = c(2.5, 0.75, 0))
##### clearly visible hump-shaped pattern in the
# partial residual plot for z|y ~ x
with(data1, {
  plot(resX1 ~ x, ylab = "z|y (partial residuals)")
  curve(coef(m1)[2] * x, min(x), max(x),
        add = TRUE, col = "blue", lwd = 2)
  mtext(side = 3, "Model 1 (wrong model)", cex = 0.7, adj = 0)
  plot(resY1 ~ y, ylab = "z|x (partial residuals)")
  curve(coef(m1)[3] * x, min(y), max(y),
        add = TRUE, col = "blue", lwd = 2)
  mtext(side = 3, "Model 1 (wrong model)", cex = 0.7, adj = 0)
})
##### no pattern in the partial residual plot for z|y ~ x
with(data1, {
  plot(resX2 ~ x, ylab = "z|y (partial residuals)")
  curve(coef(m2)[2] * x + coef(m2)[3] * x^2, min(x), max(x),
        add = TRUE, col = "blue", lwd = 2)
  mtext(side = 3, "Model 2 (correct model)", cex = 0.7, adj = 0)
  plot(resY2 ~ y, ylab = "z|x (partial residuals)")
  curve(coef(m2)[4] * x, min(y), max(y),
        add = TRUE, col = "blue", lwd = 2)
  mtext(side = 3, "Model 2 (correct model)", cex = 0.7, adj = 0)
})
```

Figure 1. Comparison of two models fitted to the same simulated data set. The dataset was simulated following the formula: $z \sim x + x^2 + y + e$, where z is the response variable, x and y are two independent predictor variables and e is the residual error (x , y and e are normally distributed with mean = 0 and sd = 1). Model 1 is misspecified as $z \sim x + y$ (only linear effects of x and y) and a clear curvature is visible in the partial residuals plot for $z|y \sim x$. Model 2 is correctly specified as $z \sim x + x^2 + y$ (linear and quadratic effect of x and linear effect of y) and the estimated polynomial relationship in the partial residual plot for $z|y \sim x + x^2$ adequately describes the pattern in $z|y$.

REVIEWERS' COMMENTS:

Reviewer #2: Editor note - The reviewer was unavailable to complete a full re-review. However, he checked your response to the question about fourth corner analysis implementation, and confirmed to us that he is satisfied with the approach and supports the idea of including both formulations of the analysis.

Reviewer #3 (Remarks to the Author):

Previous reviewer #3 here - The authors have done a great job of revising this manuscript and have addressed all of my concerns. This is a fantastic paper that will be quite impactful. Great job. I have no additional comments.

Both reviewers were satisfied with the revised version of the manuscript and our responses to their comments. Below are the reviewers' comments along with our responses (highlighted in red font).

Reviewers' comments:

Reviewer #2:

Editor note - The reviewer was unavailable to complete a full re-review. However, he checked your response to the question about fourth corner analysis implementation, and confirmed to us that he is satisfied with the approach and supports the idea of including both formulations of the analysis.

Response: The reviewer did not have additional comments.

Reviewer #3 (Remarks to the Author):

Previous reviewer #3 here - The authors have done a great job of revising this manuscript and have addressed all of my concerns. This is a fantastic paper that will be quite impactful. Great job. I have no additional comments.

Response: The reviewer did not have additional comments.